# New Discoveries on Protein Recruitment and Regulation during the Early Stages of the DNA Damage Response Pathways

**DOI:** 10.3390/ijms25031676

**Published:** 2024-01-30

**Authors:** Kelly L. Waters, Donald E. Spratt

**Affiliations:** Gustaf H. Carlson School of Chemistry and Biochemistry, Clark University, 950 Main St., Worcester, MA 01610, USA; kwaters@clarku.edu

**Keywords:** DNA damage, DNA damage response, base excision repair, nucleotide excision repair, mismatch repair, non-homologous end joining repair, homologous recombination repair

## Abstract

Maintaining genomic stability and properly repairing damaged DNA is essential to staying healthy and preserving cellular homeostasis. The five major pathways involved in repairing eukaryotic DNA include base excision repair (BER), nucleotide excision repair (NER), mismatch repair (MMR), non-homologous end joining (NHEJ), and homologous recombination (HR). When these pathways do not properly repair damaged DNA, genomic stability is compromised and can contribute to diseases such as cancer. It is essential that the causes of DNA damage and the consequent repair pathways are fully understood, yet the initial recruitment and regulation of DNA damage response proteins remains unclear. In this review, the causes of DNA damage, the various mechanisms of DNA damage repair, and the current research regarding the early steps of each major pathway were investigated.

## 1. Introduction

In 1953, Watson and Crick solved the structure of DNA, revealing that it forms a double helix [1]. This epiphany sparked further studies into DNA and identified mustard gas, alkylating agents, and UV light as sources of DNA damage in the early 1960s [2,3,4,5]. In 1964, the first DNA damage repair (DDR) mechanism, the “dark repair”, was discovered when thymine dimers were found to disappear after UV light exposure, suggesting that the cross-linked DNA was being repaired [6,7,8,9]. This mechanism is now known as the nucleotide excision repair (NER) pathway. Five years later, a defective NER pathway was identified in xeroderma pigmentosum (XP), a genetic skin disease that is associated with malignant tumors [10,11]. This discovery cemented the link between DNA damage and disease, sparking a research field focused on understanding the causes of DNA damage and the consequent repair mechanisms.

Over the past 70 years, DNA damage repair has come to be categorized based on the source causing the damage. Endogenous DNA damage occurs by internal and natural forces or processes, whereas exogenous DNA damage occurs by external or environmental factors. Exogenous and endogenous DNA damaging sources result in a variety of lesions, such as DNA adducts, DNA cross-linking, nucleotide mismatching, and DNA strand breaks [12]. When the cell senses damaged DNA, one of five common DDR mechanisms will typically be initiated—base excision repair (BER), NER, mismatch repair (MMR), homologous recombination (HR), or non-homologous end joining (NHEJ). These pathways repair damaged DNA to restore genomic stability or initiate apoptosis if the DNA cannot be repaired [13]. However, when DDR pathways fail, the DNA damage persists, causing genomic instability and mutagenesis that can lead to the proliferation of cancer cells [13].

To improve medical treatments and advance drug discovery, it is essential to more fully understand the molecular causes of DNA damage and the mechanisms involved in its repair. Despite the many years of research towards understanding repair mechanisms, there is still much to be discovered about how the cell senses DNA damage and recruits repair proteins to the damaged site. This review summarizes endogenous and exogenous DNA damage and explores current research into the mechanisms during the earlier steps of the BER, NER, MMR, HR, and NHEJ pathways.

## 2. Types of DNA Damage

### 2.1. Endogenous DNA Damage

Sources of endogenous DNA damage include spontaneous chemical reactions with nucleotides, oxidative reactions, and enzyme errors. Spontaneous reactions that cause DNA damage include hydrolysis, base deamination, and base methylation. In a hydrolytic reaction, the *N*-glycosidic bond between the sugar backbone and nitrogenous base is severed, creating an apurinic/apyrimidinic site [14,15]. Other hydrolytic reactions include the bases of adenine, guanine, and cytosine being deaminated to form hypoxanthine, xanthine, and uracil, respectively. This spontaneous reaction replaces the exocyclic amine with oxygen and occurs most frequently with cytosine [14,15]. Cytosine deamination is also catalyzed by the activation-induced deaminase/apolipoprotein B mRNA editing enzyme, catalytic polypeptide-like (AID/APOBEC) enzyme family [14,16]. Base methylation can spontaneously occur when the methyl donor *S*-adenosylmethionine (SAM, aka AdoMet) reacts with DNA. This can generate N7-methylguanine, N3-methyladenine, and O^6^-methylguanine, which can result in apurinic/apyrimidinic (AP) sites, replication inhibition, and GC → AT transitions [17].

Oxidative reactions of DNA are generally associated with mitochondrial oxidative phosphorylation. During oxidative phosphorylation, electrons can be uncoupled from the electron transport chain, which can react with oxygen to form superoxide (O_2_^–^•), a precursor to other reactive oxygen species (ROS) such as hydrogen peroxide (H_2_O_2_) and hydroxyl radical (OH•) [18]. Free H_2_O_2_ and OH• can react with DNA to cause single-stranded breaks (SSB), AP sites, and oxidized bases such as 8-oxoguanine, 5-formyluracil, and 5-hydroxycytosine [19,20,21,22]. DNA lesions can also arise endogenously due to enzyme errors. For example, despite being highly accurate, DNA polymerases can make replication errors in which base pairs can be mismatched, deleted from, or inserted into the DNA strand [23,24,25]. DNA polymerases can also slip during replication, causing the polymerase to pause and dissociate from DNA. When DNA polymerase resumes replication, it can be upstream or downstream from the site of dissociation, resulting in the insertion or deletion loops, respectively [26,27].

### 2.2. Exogenous DNA Damage

Exogenous DNA damage includes ionizing radiation (IR), ultraviolet (UV) light, and carcinogens. IR consists of high-energy X-rays, gamma rays, alpha particles, beta particles, or neutrons that can remove electrons from molecules or atoms. This radiation creates ions, radicals, and break bonds. When this energy is introduced to a cell, it causes double-stranded DNA breaks (DSB) and increased production of ROS [28,29,30,31]. Similarly, UV radiation damages DNA in the form of UVA (315–400 nm) or UVB (280–315 nm) light [32,33]. UVA light causes the production of ROS that can oxidize purine bases to produce products like 8-oxoguanine [32,33]. UVA and UVB light can also damage DNA directly by causing intrastrand-crosslinking. For example, cyclobutene pyrimidine dimers are generated from UVA and UVB lights and pyrimidine-pyrimidone (6-4) dimers can be generated from UVB light [32,33]. Carcinogens can also damage DNA in many different ways, depending on the type of carcinogen and mode of action. A well-studied carcinogen, polycyclic aromatic hydrocarbons (PAHs), has been shown to damage DNA indirectly. For example, PAHs are metabolized by Cytochrome P450 and epoxide hydrolase to generate a highly reactive diol-epoxide functional group [34,35,36]. This reactive diol-epoxide can then react with guanine and adenine bases, creating stable and bulky DNA adducts [34,35,36].

### 2.3. DDR Initiated Based on DNA Damage

Cells employ repair pathways that are specific to the different types of DNA damage detected (Figure 1). The BER pathway is well-known for correcting small DNA lesions that do not significantly distort the DNA’s helical structure, such as 8-oxoguanine, 3-methyladenine, and uracil incorporation from cytosine deamination [21,37]. Bulky lesions that significantly distort the DNA helix are repaired by the NER pathway. This includes adducts caused by PAHs, UV-induced cyclobutene pyrimidine dimers and (6-4) pyrimidine-pyrimidone dimers [14]. The MMR pathway is initiated to correct single nucleotide insertions, deletions, and mispairings, as well as insertion-deletion loops caused by DNA polymerase slipping [23,24,25,26,27]. Most DSBs are repaired by HR or NHEJ, primarily depending on the cell cycle phase. For example, NHEJ is responsible for the majority (up to 80%) of DSB repairs and is active in the G1, S, and G2 phases, while HR activity has been found to increase gradually from late G1, peak at mid-S (repairing 20–65% of DSBs), and decrease gradually towards early G2 [38]. This review seeks to highlight each pathway and the current research investigating the underlying mechanisms of protein recruitment and modes of regulation during the initial steps of the BER, NER, MMR, NHEJ, and HR pathways.

## 3. Base Excision Repair (BER)

The BER pathway is initiated by DNA glycosylases that search for lesions by 3D diffusion or by sliding along the DNA backbone via ionic interactions between the negatively charged phosphodiester backbone and positively charged amino acids in the glycosylase [39]. When a lesion is recognized, the glycosylase stops and cleaves the glycosidic bond between the damaged base and sugar, creating an AP site [40]. If the glycosylase recruited is monofunctional, an AP endonuclease then recognizes and cleaves the AP site to create 3′ OH and 5′ deoxyribose-PO_4_ ends [41]. If the glycosylase is bifunctional, the glycosylase will cleave the AP site and promote the recruitment of end processing enzymes. This allows for DNA polymerase and DNA ligase to insert the correct base pair and ligate the 3′ and 5′ ends, respectively (Figure 2).

While the mechanism of BER proteins has been extensively studied with free DNA in vitro, it is not yet clear how glycosylases overcome issues with accessing and recognizing AP lesions and small DNA base adducts in chromatin. Chromatin is made up of repeating nucleosome core particles (NCPs), with each NCP made up of about 147 base pairs that are tightly wrapped around a histone octamer [42]. If the damaged base is rotationally positioned away from the histone core or is located far from the central nucleosome dyad, a glycosylase can readily recognize and repair the lesion [43,44]. However, if a lesion is inwardly oriented and near the central nucleosome dyad, the glycosylases access to the lesion is hindered [45]. Over the past five years, researchers have investigated the effects of the NCP and histone post-translational modifications (PTMs) on DNA glycosylase activity. There are numerous glycosylases that are involved in the BER pathway that have been well reviewed and studied, including the bifunctional endonuclease III-like protein 1 (NTHL1), Nei-like protein 1 (NEIL1), and alkyladenine DNA glycosylase (AAG) [46,47,48]. Here, we focus on the recent structural and functional discoveries of two important DNA glycosylases in the BER pathway—8-oxoguanine glycosylase (OGG1) and uracil DNA glycosylase (UDG).

### 3.1. 8-Oxoguanine Glycosylase (OGG1)

OGG1 recognizes oxidized guanine bases and initiates repairs by binding to 8-oxoguanine, which can cause the DNA strand to bend and flip the damaged nucleotide out [49]. In 2017, OGG1 activity with free duplex DNA was compared to OGG1 activity at the NCP with varying rotational positioning [50]. This study found that the percentage of repaired products decreased from 87% or higher with free duplex DNA to 10% or lower with outward-facing lesions in the NCP. This result indicated that OGG1 activity is inhibited at the NCP. To further investigate the molecular mechanisms of OGG1, Olmon and Delaney modeled the solved NCP (PDB 3LZ0) and OGG1 structures (PDB 1EBM) together. Their model identified OGG1 residues R197, Y203, and R206 as potential points of steric hindrance with the histone core (Figure 3A). R197, Y203, and R206 are part of a large surface patch on OGG1 that is responsible for recognizing lesions paired with cytosine or thymine and stabilizing the DNA region opposite the lesions. The steric hindrance caused by these residues in this patch likely destabilizes the formation of an OGG1-DNA complex, thus inhibiting BER activity at an NCP [50]. To overcome these challenges at the NCP, it is suggested that chromatin remodeling, PTMs, and other proteins may be required for enhanced OGG1 activity.

In support of these findings, Bilotti et al. found that OGG1 activity fit a faster, single-exponential model for lesions on free duplex DNA and a slower, double-exponential model for lesions of the dyad at the NCP [53]. This indicates that OGG1 activity at the NCP requires both a fast and slow phase. The fast phase is attributed to the initial binding of OGG1 to the DNA. Since this study investigated lesions located at sites prone to DNA unwrapping, the slower phase could be attributed to a rate-limiting conformational change as the DNA unwraps from the NCP. This conformational change is suggested to torque and unwrap the DNA to promote the recruitment of downstream repair proteins, but the specifics of this mechanism remain unclear. To further investigate the effect of histones on OGG1 activity, the H2B *N*-terminal tail was deleted and acetylated. When the tail was deleted, the researchers observed that there was no statistically significant change in OGG1 activity. However, when the H2B tail was acetylated at various lysine residues, there was an increase in OGG1 activity. This suggests that DNA unwrapping initiated by histone acetylation promotes OGG1 repair of lesions at the NCP [53].

In addition to histone PTMs, DNA damage-related protein complexes have been suggested to assist in the recruitment and binding of OGG1 to lesions at the dyad on an NCP. For example, UV DNA damage-binding protein complex (UV-DDB) is known to contribute to the NER pathway, but recent new studies show that UV-DDB may also play a role in BER. UV-DDB is made up of the proteins DDB1 and DDB2. The UV-DDB subunit DNA damage-binding protein 2 (DDB2) has been found to assist in the activity and recruitment of OGG1 to 8-oxoguanine at telomeres [54]. It is suggested that DDB2 acts as a chromatin remodeler by mediating chromatin decompaction, thereby alleviating steric hindrance for OGG1 to promote its recruitment and binding to 8-oxoguanine at the telomere [54].

Additionally, the mediator and cohesin complexes have been identified to assist OGG1 recruitment to chromatin via an siRNA screen [55]. The mediation subunits MED12, MED14, and CDK8, as well as cohesion ring subunits SMC1, SMC3, and RAD21, were found to co-localize with OGG1 after oxidative stress [55]. Further studies about chromatin remodelers, protein recruitment, and histone PTMs can advance understanding OGG1 activity at chromatin.

### 3.2. Uracil DNA Glycosylase (UDG)

As its name suggests, enzymes within the UDG superfamily are essential for the repair of deaminated cytosine (uracil). To better investigate the roles of the UDG superfamily, human uracil *N*-glycosylase (hUNG) and its 73.3% similar homolog, *Escherichia coli* (*E. coli*) UDG, have been studied and compared [56]. Like OGG1, hUNG causes a bend in the double helix to flip out the damaged nucleotide and initiate repair [57]. UDG family members have also been observed to have issues with accessing lesions at chromatin and lesions with varying rotational positioning. For example, lesions facing towards the NCP (IN) and away from the NCP (OUT) had a much slower excision rate than lesions positioned about 90° from solution (MID) and on free duplex DNA [53]. While this study did not observe a difference in BER product formation, another study found that as the solution accessibility decreased (OUT > MID > IN), BER product formation decreased [50]. These studies suggest that UDG activity is reduced at the NCP and when the lesion is in either the MID or IN position. Interestingly, when hUNG bound to DNA (PDB 1EMH) was modeled with the NCP (PDB 3LZ0), it was found that only two residues, Y275 and R276, were within 10 Å of the histone (see Figure 3B) [50]. Since OGG1 (39 kDa) has been predicted to be sterically hindered by the histone and less active at the NCP than hUNG (35 kDa), it has been suggested that the size of the enzyme may contribute to its chromatin accessibility [50]. However, it is important to note that these are only predicted models that still need to be tested and verified using experimental approaches.

The effects of histone PTMs and histone variants have also been investigated to better understand how UDG or hUNG access lesions at the NCP. For example, the effect of H3 acetylation at K18 and K27 on UDG activity was investigated at the damaged nucleotide residue, U49 [58]. Since acetyltransferase p300 and CREB-binding protein (CBP) can acetylate the histone at K9, K14, K18, K23, and K27, this study aimed to understand if these enzymes play a role in UDG activity. UDG activity was found to increase slightly by K18 acetylation and decrease slightly by K27 acetylation when compared to activity at the canonical NCP [58]. These results suggest that p300 and CBP play a role in modulating UDG activity. However, with only one site of damage and two sites of acetylation investigated, the role of p300 and CBP cannot be conclusively determined at this time and requires further investigation. Various positions of nucleotide damage and other acetylation targets, particularly acetylation at K9, K14, and K23, still need to be examined to clarify if p300 and CBP influence UDG activity.

Another study conducted by Li and Delaney investigated the effect of histone variants on UDG activity. In their 2019 study, the H2A variants macroH2A and H2A.Z were found to generally enhance UDG at the NCP [59]. UDG activity at the dyad was found to significantly increase with the macroH2A NCP [59]. UDG excision of lesions with poor solution accessibility were also significantly increased with the H2A.Z and macroH2A NCPs [59]. In their follow-up 2022 study, Li et al. observed that the histone variant H3.3 NCP and double-variant H2A.Z/H3.3 NCP increased U excision via UDG at lesions with IN and MID rotational positioning [60]. While both variants increased UDG activity at the dyad compared to the canonical NCP, the H3.3 NCP was found to increase U excision more than H2A.Z/H3.3 at the dyad. These studies suggest that the histone variants H2A.Z, macroH2A, H2A.Z/H3.3, and H3.3 alter the structure and dynamics of DNA wrapping on NCPs thereby allowing for increased UDG efficiency compared to canonical NCP [60]. Further structural and biochemical studies involving histone PTMs, chromatin remodelers, and other histone variants would help clarify UDG activity at the NCP.

## 4. Nucleotide Excision Repair (NER)

DNA damage repair proteins recognize bulky lesions to initiate the transcription coupled-NER (TC-NER) or the global genome-NER (GG-NER) pathways (Figure 4). In the TC-NER pathway, RNA polymerase II stalls at the DNA lesion site and recruits Cockayne syndrome group B (CSB), Cockayne syndrome group A (CSA), and UV-stimulated scaffold protein A (UVSSA) [61]. This leads to the recruitment of transcription factor IIH (TFIIH). Alternatively, in the GG-NER pathway, most lesions are recognized by a multiprotein complex that contains XP group C-complementing protein (XPC), radiation sensitive 23B (RAD23B), and Centrin 2 (CETN2) [62]. However, UV-induced pyrimidine dimers are more readily recognized by the UV-DDB complex that recruits XPC, where DDB2 binds to the damaged site and flips two bases into the recognition pocket [63,64]. DDB1 functions as a substrate for the E3 ubiquitin ligase complex CUL4-RBX1 that can ubiquitylate DDB2 and XPC [65]. When DDB2 is ubiquitylated, its binding affinity towards DNA is reduced, and DDB2 is targeted for proteasomal degradation, thus disrupting the GG-NER pathway [66]. However, when the XPC-RAD23B-CETN2 complex is recruited, the DDB1-CUL4-RBX1 E3 ligase complex will preferentially ubiquitylate XPC, leading to the recruitment of TFIIH [66].

The function and structure of TFIIH have been extensively studied [67,68]. TFIIH is a giant 0.5 MDa complex with 10 subunits—general TFIIH subunit 1 (GTF2H1 or p62), GTF2H2 (p44), GTF2H3 (p34), GTF2H4 (p52), GTF2H5 (p8), XP group B-complementing protein (XPB), XP group D-complementing protein (XPD), RING finger protein MAT1, cyclin H, and cyclin-dependent kinase 7 (CDK7) [67,68]. In both the TC-NER and GG-NER pathways, TFIIH recruits XPA and RPA to unwind the DNA helix and form a bubble in the DNA. The excision repair cross-complementation group 1-XP group F-complementing protein (ERCC1-XPF) complex and XP group G-complementing protein (XPG) are then recruited to cut the DNA strand 5′ and 3′ of the lesion, respectively [62]. This allows for DNA polymerase and DNA ligase to fill and repair the damaged site. Despite being the first discovered repair pathway, the detailed molecular mechanisms for NER protein recruitment and regulation remain unclear and require further investigation.

### 4.1. Transcription-Coupled Nucleotide Excision Repair (TC-NER)

Weegen et al. examined the recruitment of CSB, CSA, UVSSA, and TFIIH to DNA damage-stalled RNA polymerase II using human knockout cells and a unique immunoprecipitation method to isolate stalled RNA polymerase II [69]. CSB was found to be the first protein recruited to stalled RNA polymerase II. This binding is thought to bring about a conformational change in CSB that exposes the CSB *C*-terminal CSA-interaction motif (CIM) to promote CSA recruitment. UVSSA then uses its Vps-27, Hrs, and STAM (VHS) domain to interact with CSA while being stabilized by CSB. TFIIH is then recruited to RNA polymerase II by using subunit p62 to interact directly with UVSSA residues 400–500 [69,70]. Since the downstream repair proteins, XPA and ERCC1-XPF, were not detected with the RNA polymerase pulldowns, it is hypothesized that the RNA polymerase II and CSB-CSA-UVSSA complex may be shifted or possibly removed when TFIIH binds with XPA [69]. In support of these findings, Mistry et al. found that there is a large structural change in UVSSA upon TFIIH binding that causes UVSSA to become more compact and may cause the dissociation of the CBS-CBA-UVSSA complex from DNA and/or RNA Polymerase II [71].

Weegen et al. also detected the DDB1-CUL4-RBX1 E3 ligase complex in their pulldown assays, suggesting that the E3 ligase complex binds to CSA [69]. This supports findings from other studies that have found that DDB1-CUL4-RBX1 binds to CSA via DDB1 to ubiquitylate RNA polymerase II at K1268, UVSSA at K414, and potentially CSB at K991, K1392, and K1457 [69,72,73,74,75]. Ubiquitylation of RNA polymerase II and UVSSA was found to promote interactions with TFIIH, and the ubiquitylation of CSB was found to play a regulatory role by targeting CSB for proteasomal degradation [69,72,73,74,75]. The deubiquitinating enzyme USP7 has been reported to bind to UVSSA to stabilize the interaction between UVSSA and CSB and regulate CSB deubiquitylation [70,76,77,78]. While these studies have provided great insight into the mechanisms of TC-NER protein recruitment, the regulation and structural basis for the assembly of these ubiquitylation-regulating enzymes and the downstream recruitment of the TFIIH complex remains unclear.

### 4.2. Global Genome-Nucleotide Excision Repair (GG-NER)

The role of histone deacetylases (HDACs) has been investigated to better understand GG-NER regulation and protein recruitment. For example, HDAC1, HDAC2, and HDAC activators (metastatic-associated proteins) have been found to promote DNA damage recognition and recruitment of XPC in *Ddb2*-deficient cells [79]. This study found that XPC preferentially localizes to hypoacetylated chromatin and interacts with deacetylated H3 tails via a disordered region in XPC (residues 325–512) [79]. Supporting these findings, HDAC3 has also been found to deacetylate histone H3 residue K14 after UV irradiation, as well as promote XPC recruitment to DNA [80]. Interestingly, ubiquitylation of XPC by DDB1-CUL4-RBX1 was delayed with HDAC3 depletion, indicating that H3 K14 deacetylation is important to promote XPC ubiquitylation. Further functional and structural studies are needed to fully understand the role of H3 dynamics in the recruitment and ubiquitylation of XPC during GG-NER.

To decipher the regulatory role of ubiquitin in GG-NER, the effect of the deubiquitylating enzyme USP44 on XPC and DDB2 has been investigated by Y. Zhang et al. In their 2021 study, DDB2 and XPC recruitment to damaged DNA was diminished in *Usp44*-deficient murine embryonic fibroblasts [81]. The researchers also found that *Usp44*-deficient mice have a significant increase in skin tumor formation caused by UVB-induced DNA damage. This indicates that USP44 may regulate the GG-NER pathway by deubiquitylating DDB2 to prevent DDB2-proteasomal degradation and promote the recruitment of XPC [81]. To uncover the precise role that deubiquitylating enzymes play in GG-NER regulation, the mechanisms used by USP44 to deubiquitylates DDB2 require further investigation.

## 5. Mismatch Repair (MMR)

The MMR pathway is initiated when DNA polymerase errors during replication that cause nucleotide mismatching or insertion-deletion loops are detected by the cell machinery. For example, mismatched nucleotides and small insertion-deletion loops (1–4 nucleotides) are recognized by the heterodimeric MutSα complex comprised of MutS homolog 2 (MSH2) and MutS homolog 6 (MSH6) [14,82,83,84]. Alternatively, longer insertion-deletion loops (3 nucleotides or larger) are recognized by the MSH2 and MutS homolog 3 (MSH3) heterodimer, MutSβ [14,82,83,84]. The MutSα and MutSβ complexes bind to DNA by interacting with the helix primarily with the MSH6 and MSH3 subunits, respectively [85,86]. Once bound to DNA, these MutS complexes are thought to hydrolyze ATP to form a sliding clamp that searches for the damaged DNA sites, but the precise mechanisms used by these complexes remain unclear [87,88,89,90].

When a lesion is recognized, the MSH complex stops and recruits the MutLα heterodimer consisting of MutL homolog 1 (MLH1) and post-meiotic segregation increased 2 (PMS2) [91]. Replication factor C (RFC) is also recruited to aid in recruiting proliferating cell nuclear antigen (PCNA) to the DNA and possibly activate MutLα endonuclease activity [82,83,88]. Endonuclease EXO1 can then be recruited to the damaged DNA site and assist in the removal of damaged DNA [82,83,88]. DNA Polymerase δ, β, and ƞ are then recruited to fill the gap, and DNA ligase seals the nick (Figure 5) [82]. The precise mechanisms and interactions of the early steps in the MMR pathway are unclear and require further functional and structural analysis.

### 5.1. MutS Mechanisms and Function

While MutS has been found to undergo a conformational change and multiple states while transitioning from ADP to ATP, there has not yet been a detailed structural description of these dynamic processes [87,88,89,90,92,93,94,95,96,97]. Recent cryo-electron microscopy and X-ray crystallography structures of the *E. coli* homodimeric MutS provide new clarity on how the MutS homodimer uses ATP hydrolysis to transform into a sliding clamp [98,99]. When MutS initially binds to DNA, ATP hydrolysis is prevented, and the homodimer is held in an open conformation to search for the damaged DNA [99] (Figure 6A). When a lesion is found, MutS undergoes a structural rearrangement that enables ATP hydrolysis and forms the sliding clamp by closing the MutS dimer [98,99] (Figure 6B). This causes a bend in the DNA and promotes MutL to bind to the damaged DNA site to initiate the MMR [98,100] (Figure 6C,D). While these structural studies provided valuable insight into the possible MMR mechanisms of the homodimer MutS in *E. coli*, the human heterodimers MutSα and MutSβ may not regulate MER activity using the same mechanism as bacterial MMR.

Recently, many differences in human and bacterial MutS dimers have indicated that human MutS complexes have a different function and mechanisms of action. For example, human MutSα has been found to bind to DNA in a two-state model instead of the 1:1 binding model that *E. coli* MutS fits [101,102]. Human MutSα and MutSβ were also found to have a longer lifetime as a sliding clamp than *E. coli* MutS, which suggests that the human MutS complexes may have a different mechanism of action than bacterial MutS [103]. Human MutSα functional studies have also revealed a novel role of human MutS complexes where MutSα sliding is required for EXO1 removal of the damaged base and not for initiating MMR [104]. In this mechanism, MutSα binding to DNA promotes MutLα recruitment, which cleaves the 5′ end of the mismatched daughter strand. As EXO1 is recruited to the 5′ nicked DNA, the MutSα-MutLα complex slides away in an ATP-dependent manner to allow EXO1 to cut the 3′ end and remove the damaged base [104]. While these studies have advanced the current understanding of various MutS complex functions in humans and *E. coli*, the mechanisms underlying these functions are not yet clear for human MutSα and MutSβ.

### 5.2. MutLα Regulation by PTMs

MLH1 is an important component of the MutLα complex that has been found to undergo various PTMs to regulate the MMR response. Recently, MLH1 has been found to be acetylated at residues K33, K241, K361, and K377 [105]. In the same study, HDAC6 was found to deacetylate MLH1 in vitro and in vivo. Deacetylated MLH1 was shown to inhibit the assembly of the MutSα-MutLα complex, suggesting that MLH1 acetylation is essential to regulate MMR activity. MLH1 has also been found to be phosphorylated at S87, S446, S456, and S477 [106]. S87, located in the ATPase site of MLH1, was shown to inhibit MutLα DNA binding and disrupt MMR activity when phosphorylated. Additionally, phosphorylation of S446, S456, and S477, all located in the linker region of MLH1, were found to inhibit MMR activity. This study suggests that MLH1 phosphorylation is essential to regulate MutLα activity and the MutLα interaction with MutSα. Further studies are needed to better understand how PTMs affect MLH1 function and MMR regulation.

## 6. Non-Homologous End Joining (NHEJ)

NHEJ is the most common repair pathway for mutagenic DSBs and requires the coordination of many different proteins. NHEJ repair is initiated when two Ku70-Ku80 ring-shaped heterodimers bind adjacent to the DSB on each strand [107] (Figure 7). Ku serves as a loading protein to recruit other NHEJ proteins to the site of DNA. DNA-dependent protein kinase catalytic subunit (DNA-PKcs), a phosphatidylinositol 3-kinase-related kinase (PIKK) family member, is then recruited to the Ku70-Ku80 complex. Upon binding to DNA and the Ku70-Ku-80 complex, DNA-PKcs interacts with another DNA-PKcs complex on the other DNA strand. This interaction pulls the broken strands closer together by pushing the Ku70-Ku80 dimers further from the DSB, activating the DNA-PKcs serine/threonine kinase activity [108,109,110] (Figure 7). DNA-PKcs phosphorylates various NHEJ proteins during this process, including Ku70 (at S6), Ku80 (at S577, S580, and T715), itself, X-ray repair cross-complementing protein 4 (XRCC4; at S320), and Artemis (at S516 and S645) [14,111,112,113,114,115].

To prepare the DNA strands for ligation, end-processing enzymes such as Artemis, polynucleotide kinase/phosphatase (PNKP), tyrosyl-DNA phosphodiesterase 1 and 2 (TDP1 and TDP2), Aprataxin, and polymerases pol µ, pol λ, and terminal deoxynucleotidyl transferase (TdT) are recruited [113,116,117]. Artemis cleaves hairpin turns and removes 5′-aldehyde structures after DNA-PKcs are recruited [116]. TDP1, TDP2, PNKP, and Aprataxin remove 5′ or 3′ adducts and restore the 3′ hydroxyl and 5′ phosphate ends of the broken DNA [116]. Polymerases pol µ, pol λ, and TdT can fill in gaps and overhangs [116,118]. While Artemis is known to contribute to the NHEJ repair after DNA-PKcs, the sequence of events that causes other end-processing enzymes to be recruited remains unclear.

To complete the NHEJ repair, Ligase IV, XRCC4, XRCC4-like factor (XLF), and Paralog of XRCC4 and XLF (PAXX) are recruited [119,120]. XRCC4, Ligase IV, and XLF form a bridge with the Ku70-Ku80 complexes on both ends of the DSB. The XRCC4-Ligase IV complex interacts with the Ku70-Ku80 and DNA-PKcs complex on each DNA strand [119,120]. XLF forms a β-strand zipper via hydrogen bonds with both XRCC4 proteins to create a scaffold that is stabilized by PAXX interactions with the Ku70-Ku80 complex [120]. This pulls the DNA strands closer together and allows for Ligase IV to mend the broken DNA (Figure 7) [113,116,117,120].

### 6.1. Structural Characterization of NHEJ Complexes

Over the past 3 years, the understanding of NHEJ mechanisms has vastly improved due to the significant amount of NHEJ complex structures that have been solved. In a recent excellent review by Vogt and He, the alternative mechanisms for NHEJ were discussed in great detail based on the solved structures of NHEJ complexes [121]. Future structural work on the recruitment and mechanisms of end-processing enzymes will help to further clarify how NHEJ is controlled.

### 6.2. Regulation of DNA-PKcs via PTMs

Studies have investigated the effect of various PTMs to better understand how DNA-PKcs function is regulated. For example, DNA-PKcs have been found to be acetylated at K3241 and K3260 to promote genomic stability at basal conditions [122]. However, when exposed to IR, SIRT2 has been found to deacetylate DNA-PKcs and promote the localization of DNA-PKcs to DSBs and the Ku70-Ku80 complex [123]. This suggests that SIRT2 contributes to NHEJ by sensing the DNA damage and activating DNA-PKcs.

Ataxia telangiectasia-mutated (ATM), a PIKK family member, has also been implicated in the NHEJ repair by phosphorylating DNA-PKcs at T4102 [124]. Pulldown assays and fluorescent-tagged real-time recruitment assays suggest that T4102 phosphorylation stabilizes the DNA-PKcs and Ku70-Ku80 complex and promotes the recruitment of XRCC4, XLF, and PNKP [124]. While there is no structural data currently available, it is hypothesized that DNA-PKcs-dependent phosphorylation of ATM at T4102 induces a conformational change that results in a more stable form of the DNA-PKcs and Ku70-Ku80 complex. This more stable complex is suggested to promote NER activity by enhancing the recruitment of other NHEJ repair proteins.

### 6.3. Regulation of DNA-PKcs via Autophosphorylation

DNA-PKcs kinase activity has been found to depend on whether the DSB has a hairpin, blunt, or overhanging end [125]. Hairpin and 5′ overhanging ends were found to promote DNA-PKcs *cis*-autophosphorylation of T2609, T2638, T2645, and T2647 in the ABCDE patch (amino acids 2580–2780) and the recruitment of Artemis. This phosphorylation event opens up the DNA-PKcs, allowing Artemis to cleave the hairpin or 5′ overhanging end [125]. Blunt and 3′ overhanging ends were found to be protected by DNA-PKcs by inhibiting DNA-PKcs *cis*-autophosphorylation and likely promoting the *trans*-autophosphorylation of the PQR patch (amino acids 2000–2060) and phosphorylation of other targets [125]. The functional significance of PQR phosphorylation is not yet known, but knockout studies suggest that PQR phosphorylation and XLF may play a cooperative role in NHEJ [126]. While the role of DNA-PKcs autophosphorylation has become clearer, further studies are needed to decipher how DNA-PKcs phosphorylation contributes to the recruitment of other NHEJ proteins.

## 7. Homologous Recombination (HR)

In contrast to HR, NHEJ does not require a template strand and occurs faster and more frequently in the cell. Due to this tendency, NHEJ can result in sequence errors and is considered to be less accurate than HR. HR repairs have a high fidelity because HR uses a sister chromatid as a template to repair the DSB. Because a sister chromatid is required, this repair pathway typically occurs in the S or G_2_ phases of the cell cycle [127]. Generally, HR is initiated with the recognition of a DSB and consequent resection of the 5′ ends of the damaged DNA. The overhanging 3′ end of the damaged DNA invades a sister chromatid and uses the complementary sequence to elongate the damaged DNA, forming a D-loop [128,129]. The DNA will then be repaired by the synthesis-dependent strand annealing (SDSA) pathway or by forming a double Holliday junction [128]. For SDSA, the D-loop is disrupted and annealed to the other broken strand. This allows DNA polymerases and ligases to be recruited, fill in nucleotide gaps, and seal the nicked DNA (Figure 8A) [128,129]. Alternatively, a double Holliday junction is formed when the D-loop is disrupted by another end of the damaged DNA strand invading the sister chromatid. The Holliday junction allows both strands to be elongated before annealing with a complementary DNA strand. This can result in crossover between sister chromatids or non-crossover once fully repaired (Figure 8B) [124,128,129]. For HR to occur, many proteins and protein complexes are required for each step. The following sections will discuss the proposed mechanisms and research regarding DSB recognition and HR excision.

### 7.1. DSB Recognition and the MRE11-Rad50-NBS1 Complex

In response to DSBs, c-Jun *N*-terminal kinase (JNK) phosphorylates S10 of SIRT6 to recruit poly(ADP-ribose) polymer 1 (PARP1) to the damaged site [130,131]. This allows for the poly(ADP-ribosyl)ation (PARylation) of histones and the recruitment of chromatin remodeler Alc1 to PARP1 that promotes the recruitment of the MRE11-Rad50-NBS1 (MRN) complex to the DSB [132,133]. Upon sensing the DSB, ATM is thought to be recruited by MRN in its inactive dimer form, as shown in a recently solved structure of the ATM dimer bound to a NBS1 peptide [134] (Figure 9A). When localized to the DSB, ATM is activated by dissociating into a monomer via autophosphorylation at S1981 [135]. This allows ATM to phosphorylate S139 of H2AX (γH2AX) and promote H2AX-binding with a mediator of DNA damage checkpoint 1 (MDC1) via the breast cancer-associated *C*-terminal domain as shown via X-ray diffraction [136,137] (Figure 9B). Phosphorylation of the MDC1 SDT repeats by casein kinase 2 (CK2), which is thought to promote an interaction between MDC1 and NBS1, a component of the MRN complex. This MDC1-NBS1 interaction is predicted to stabilize the MRN-ATM complex when bound to DNA [135,138,139]. 

After the MRN complex binds to the damaged DNA, ATP hydrolysis via the Rad50 subunit is thought to destabilize the double-helix, causing the ends to partially unwind [141]. A recently solved structure of the *Chaetomium thermophilum* MRN complex bound to ATPγS has provided valuable insight into MRN tethering of DNA strands and catalytic activity, but this has yet to be solved for the human complex (Figure 9C) [140]. MRN endonuclease activity is driven by the protein cofactor *C*-terminal binding protein (CtBP)-interacting protein (CtIP). CtIP is phosphorylated at T847 by cyclin-dependent kinases during the S phase of the cell cycle and, therefore, is thought to be a key component to initiating HR and not NHEJ [142,143]. Phosphorylated CtIP promotes endonuclease activity by binding to the forkhead-associated (FHA) domain of NBS1 [142,144,145]. Recently, solved and predicted structures of CtIP domains have been modeled to reveal dimeric (PDB 7BGF), and tetrameric (PDB 4D2H) conformations of CtIP, but the functional aspects and overall structure of CtIP have yet to be solved [146,147,148].

### 7.2. Targeting Proteins to the DSB

New studies have helped reveal the molecular mechanisms used to recruit other proteins to DSB sites. For example, the breast cancer type 1 susceptibility protein (BRCA1) *C*-terminal (BRCT) domain of MDC1 binds to phosphorylated Ser139 of λH2AX [137]. MDC1 is further phosphorylated by ATM, which promotes the recruitment of the E3 ubiquitin ligase RING finger protein 8 (RNF8) and the E2 conjugating enzyme UBC13 [149,150]. This leads to the polyubiquitination of H1-type linker histones, which is essential for recruiting another E3 ubiquitin ligase RING finger protein 168 (RNF168) to the complex [151]. RNF168 then monoubiquitinates K13 and K15 of H2A and H2AX, and RNF8 is responsible for extending the ubiquitin chains, resulting in the polyubiquitylation of H2A and H2AX [152]. BRCA1 and BRCA1-associated RING domain protein 1 (BARD1) are recruited to mono-ubiquitylated H2A via receptor-associated protein 80 (RAP80) that promotes the excision of the 5′ ends and recruitment of replication protein A (RPA) [153,154,155,156,157,158,159]. While its functional relevance is not yet clear, another E3 ligase, homologous to the E6AP *C*-terminus (HECT) and RCC1-like domain (RLD)-containing protein 2 (HERC2) has been implicated in the regulation of HR via an interaction with RNF8 and ubiquitylation of BRCA1 [160,161,162,163,164]. Further studies that characterize the recruitment of BRCA1 and other associated proteins are needed during HR are needed.

Once the 5′ ends of the damaged DNA are resected, RPA rapidly binds to the 3′ overhanging ends to protect the DNA by preventing other DNA structures from forming until it is ready to invade the sister chromatid [165]. The partner and localizer of BRCA2 (PALB2) are recruited to the BARD1-BRCA1 complex, which leads to the recruitment of BRCA2 and Rad54 [166,167,168,169]. This promotes the replacement of RPA with RAD51, which leads to sister strand invasion and consequent repair of the damaged DNA [168,170]. While the pathway of HR has become clearer, there is still a lack of solved structures that demonstrate the precise mechanisms and structural changes required to initiate HR in humans. Further studies regarding the regulation and structural impacts of protein-protein interactions are required to advance our understanding of the molecular basis for HR regulation.

## 8. Conclusions

In this review, we have discussed the causes of DNA damage and the current models for initiating the DDR response. While our understanding of BER, NER, MMR, NHEJ, and HR general pathways and mechanisms has greatly improved, there is still much to learn about the precise mechanisms for how and why a DDR pathway is activated. Investigating the mechanisms of repair protein recruitment, the conformational changes upon repair protein interactions, and how repair responses are regulated by PTMs or chromatin remodelers is needed. These studies will further our understanding of DNA damage responses and, therefore, aid in developing therapeutic targets for diseases such as cancer and Huntington’s disease. While our understanding of DNA damage responses has greatly advanced over the past 50 years, further research is needed to understand the precise mechanisms and regulation of the eukaryotic DNA repair responses.

## Figures and Tables

**Figure 1 ijms-25-01676-f001:**
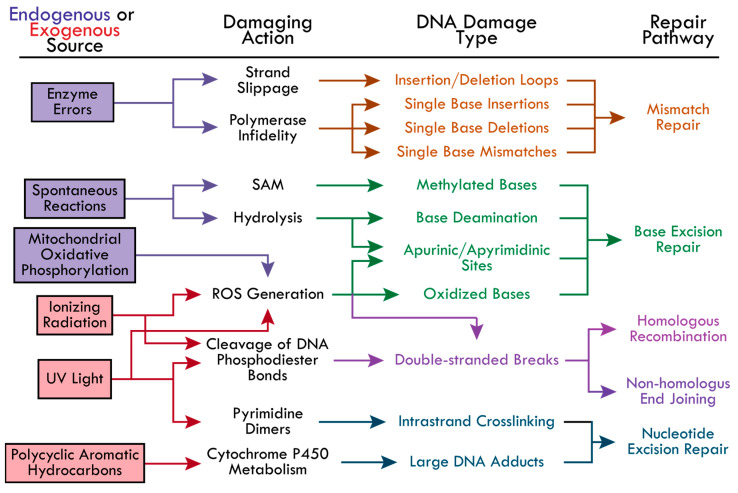
The causes of DNA damage and the corresponding repair pathways. The possible DNA-damaging actions from endogenous (boxed in purple) and exogenous (boxed in red) sources are indicated by colored-coordinated arrows. The types of DNA damage resulting from each damaging action and the corresponding repair pathways are colored accordingly—Mismatch repair (MR, orange), base excision repair (BER, green), homologous recombination (HR, pink), non-homologous end joining (NHEJ, pink-purple), and nucleotide excision repair (NER, blue). The following abbreviations were made in the figure: reactive oxygen species (ROS), *S*-adenosylmethionine (SAM), cyclobutene pyrimidine dimers and (6-4) pyrimidine-pyrimidone dimers (Pyrimidine Dimers).

**Figure 2 ijms-25-01676-f002:**
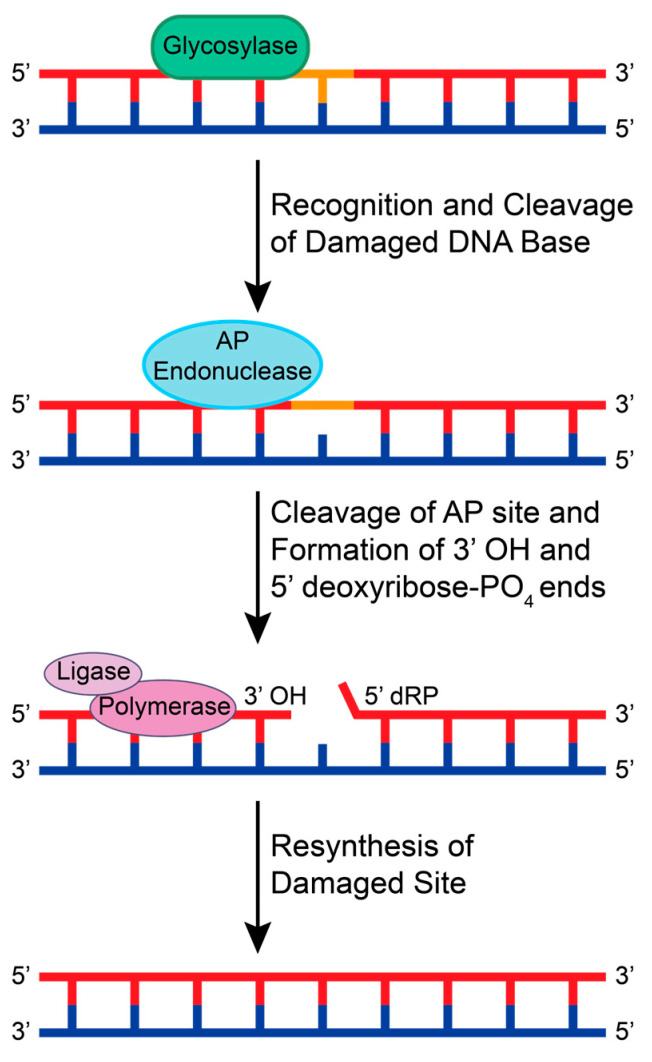
A schematic of the general BER pathway using a monofunctional glycosylase. A glycosylase recognizes a small adduct and cleaves the damaged base from the sugar backbone, creating an AP site. An AP endonuclease recognizes and cleaves the AP site to create 3′ OH and 5′ deoxyribose-PO_4_ (dRP) ends. Generally, DNA polymerases and DNA ligases are recruited to resynthesize and repair the damaged base.

**Figure 3 ijms-25-01676-f003:**
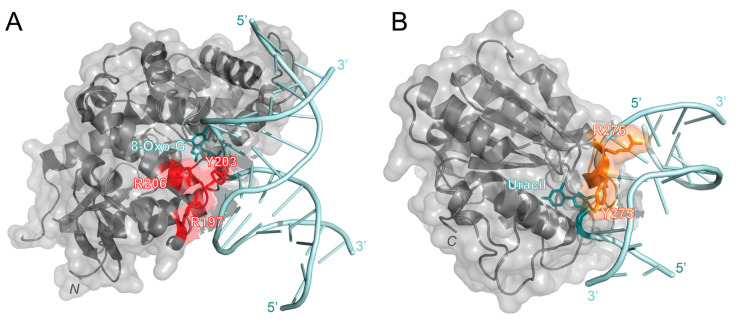
Predicted interaction sites of OGG1 (PDB 1EBM) and hUNG (PDB 1EMH) to DNA in an NCP [49,51]. (**A**) The structure of OGG1 (gray) is bound to the 8-oxoguanine lesion (cyan) in a DNA strand (light blue). The predicted sites of steric clashes with the histone are colored red (R197, Y203, and R206). (**B**) The structure of human uracil *N*-glycosylase (hUNG, gray) bound to uracil (cyan) in a DNA strand (light blue). Residues that are predicted to be within 10 Å of the histone are colored orange (Y275 and R276). Structures were visualized with PyMOL [52].

**Figure 4 ijms-25-01676-f004:**
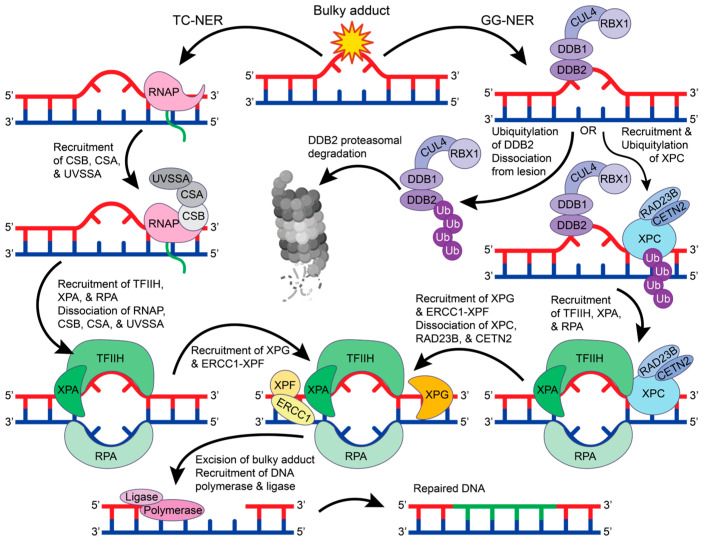
The GG-NER and TC-NER pathways. In TC-NER (left), RNA polymerase (RNAP) stalls at the lesion to recruit CSB, CSA, and UVSSA, which recruits TFIIH, XPA, and RPA. Upon TFIIH, XPA, and RPA binding, the RNAP complex dissociates from the DNA. In GG-NER, the lesion is recognized by the DDB2-DDB1-CUL4-RBX1 E3 ubiquitin ligase complex that will either ubiquitylate DDB2 for proteasomal degradation or recruit XPC-RAD23B-CETN2. Ubiquitylation of XPC leads to the recruitment of TFIIH, XPA, and RPA. In both TC-NER and GG-NER, TFIIH, XPA, and RPA create a bubble in the double helix to allow for the recruitment of XPG and ERCC1-XPF. In GG-NER, when XPG and ERCC1-XPF are recruited, the XPC-RAD23B-CETN2 complex dissociates from the DNA. XPG and ERCC1-XPF excise the bulky DNA lesion, which can then be repaired by DNA polymerase and DNA ligase. The figure was made in Adobe Illustrator with the proteasome supplemented from BioRender.com (accessed on 9 October 2023).

**Figure 5 ijms-25-01676-f005:**
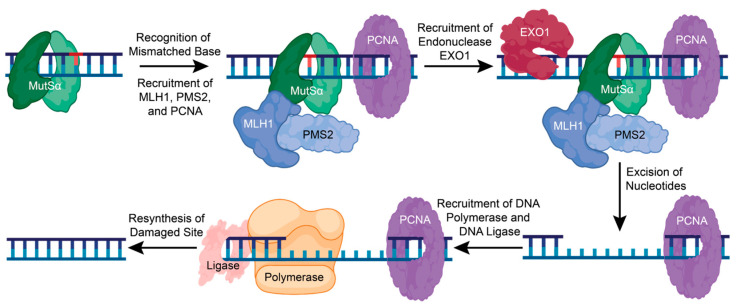
The mismatch repair pathway corrects a single nucleotide error. In MMR, a single mismatched, inserted, or deleted nucleotide is recognized by the sliding clamp complex, MutSα. This promotes the recruitment of MLH1, PMS2, and PCNA, which then recruits the endonuclease, EXO1. EXO1 removes the damaged base pair and other nucleotides to recruit DNA polymerase and DNA ligase to resynthesize and repair the damaged DNA. Figure created with BioRender.com (accessed on 16 January 2024).

**Figure 6 ijms-25-01676-f006:**
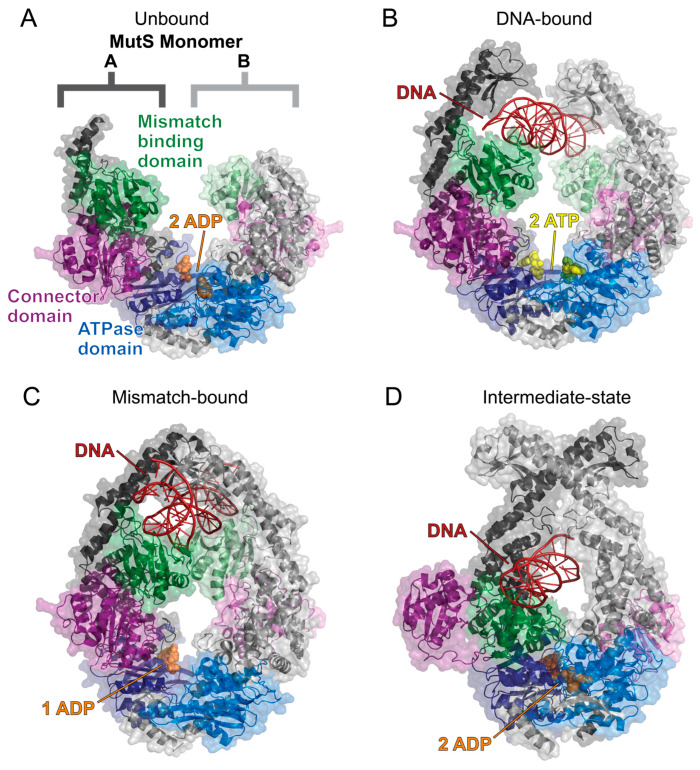
The solved cryo-EM structures of the *E. coli* MutS homodimer sliding clamp. The MutS dimer is labeled and colored such that monomer A has a darker tone than monomer B. All structures are colored as follows: mismatch binding domain (residues 2–115, green), connector domain (residues 116–266, purple), ATPase domain (residues 568–765, blue), ADP (orange spheres), ATP (yellow spheres), and DNA (red). (**A**) The MutS homodimer in a complex with two ADP molecules (PDB 7OU2) is flexible and could not be fully resolved using cryo-EM [99]. (**B**) The sliding clamp is formed when MutS binds to DNA and two ATP molecules (PDB 7AI5) [98]. This allows for the mismatch binding domains to search for damaged DNA. (**C**) When DNA damage is recognized, the mismatch binding domains bind to the DNA strand, and an ATPase domain catalyzes ATP hydrolysis to induce a bend in the DNA as the MutS sliding clamp closes around the DNA strand (PDB 7AI6) [98]. (**D**) ATP hydrolysis at both ATPase domains causes a large conformational change, as shown by the MutS dimer in a complex with two ADP molecules and DNA (PDB 7AI7) [98]. The MutS dimer closes tighter around the DNA strand, which forces the connector domains (purple) and mismatched binding domains (green) to flip out. Structures were visualized with PyMOL [52].

**Figure 7 ijms-25-01676-f007:**
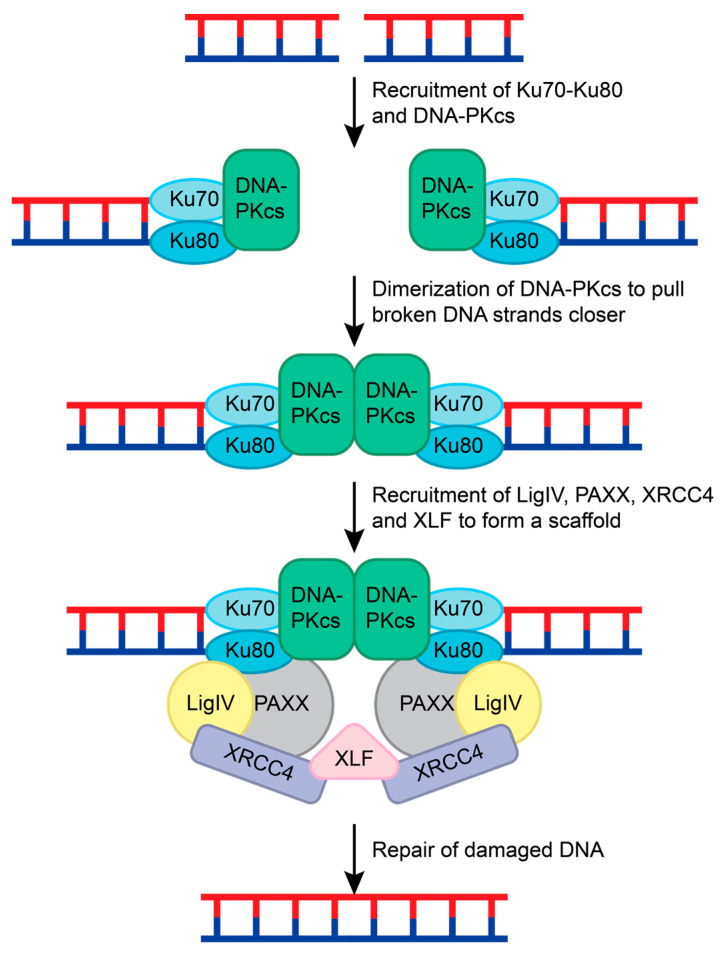
NHEJ repair of double-stranded breaks. The Ku70-Ku-80 complex recognizes the double-stranded break and recruits DNA-PKcs to both ends of the damaged DNA. DNA-PKcs form a dimer and pull the broken strands closer together. This leads to the recruitment of LigIV, PAXX, and XRCC4 to both DNA strands. This complex is held together by XLF, and the DNA strands are ligated together to repair the double-stranded break.

**Figure 8 ijms-25-01676-f008:**
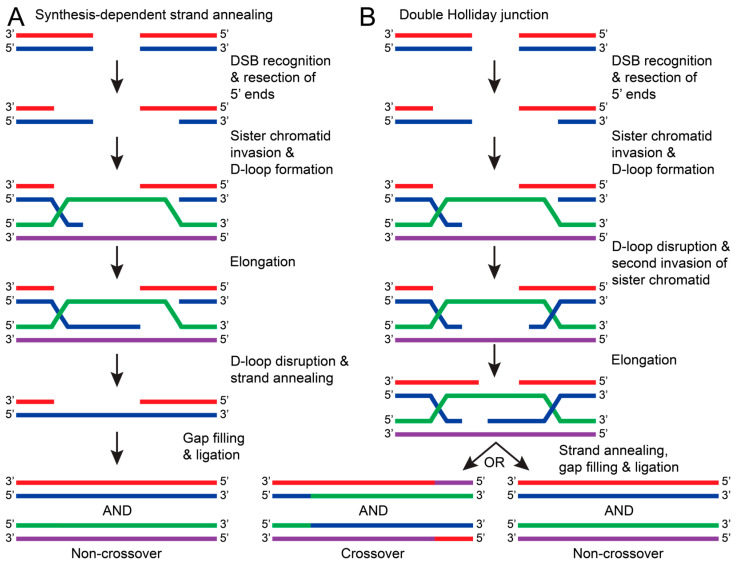
The general pathways of DNA repair via HR. In both pathways, HR is initiated by recognizing DSBs and resecting the 5′ ends of the damaged DNA strand (red with blue). The overhanging 3′ end of the damaged DNA (blue) invades the sister chromatid (green with purple) to form a D-loop. (**A**) In synthesis-dependent strand annealing (SDSA), the damaged DNA strand (blue) is elongated using a sister chromatid DNA strand as a template (purple). The D-loop is disrupted, causing the damaged DNA strands to anneal (red with blue). DNA polymerase and ligases fill in nucleotide gaps and ligate the stand to result in a non-crossover repair (red with blue and green with purple). (**B**) If the D-loop is disrupted by a second invasion of the damaged strand (blue), then a double Holliday junction is formed. Both damaged DNA strands (red and blue) are elongated, but when the junction resolves, both crossover (red/purple with blue/green and green/blue with purple/red) and non-crossover products can occur.

**Figure 9 ijms-25-01676-f009:**
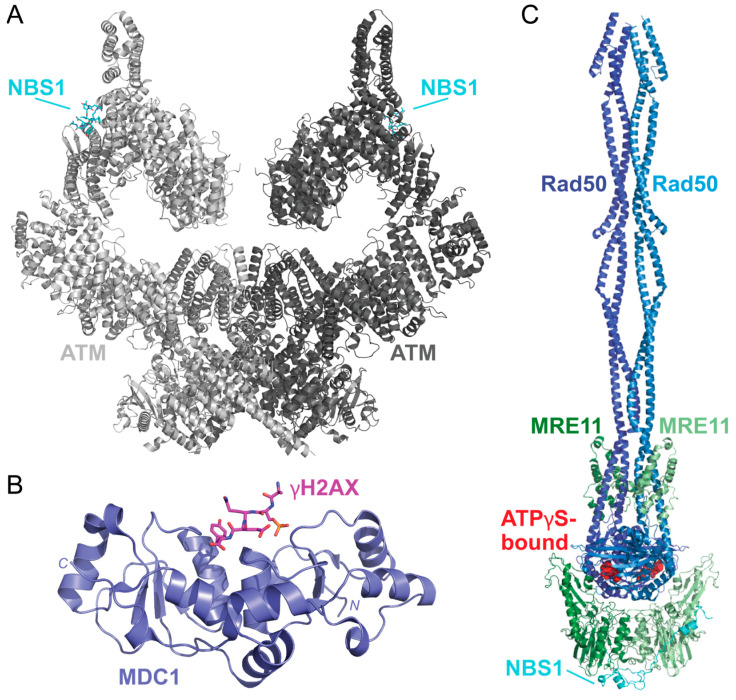
The solved structures of DSB response proteins ATM, MDC1, and MRN. (**A**) The structure of the ATM (shades of gray) bound to NBS1 (cyan) solved via cryo-EM shows that there are two possible NBS1-binding sites on the ATM dimer (PDB 7SID) [134]. Each ATM monomer is shown in a different shade of gray. (**B**) The structure of the BRCT domain of MDC1 (purple) bound to a γH2AX phosphopeptide (pink) was solved using X-ray crystallography (PDB 2AZM) [137]. The γH2AX peptide contains the *C*-terminal residues 134–143 and a phosphoserine at residue S140 (KKATQA-pS-QEY). (**C**) The solved structures of the MRN complex from *Chaetomium thermophilum* are bound to ATPγS (red spheres). This structure shows how MRE11 (shades of green) and Rad50 (shades of blue) form a dimer that is scaffolded together by NBS1 (cyan). This complex was solved using cryo-EM (PDB 7ZR1) [140]. Structures were visualized using PyMOL [52].

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
