# Peer review of "New Discoveries on Protein Recruitment and Regulation during the Early Stages of the DNA Damage Response Pathways"

_ijms, 2024, doi:10.3390/ijms25031676_

Round 1
Reviewer 1 Report
Comments and Suggestions for Authors
The authors review the repair pathways and mechanisms involved in the DNA damage response. The review does not go into great detail on these subjects, but does provide a reasonable level of depth for some topics.
Therefore, I have suggestions to improve this manuscript.
The title of the manuscript “The Repair Pathways and Mechanisms involved in the DNA Damage Response” is too global if elucidate the information inside the review. This affected title may be an attempt to disguise the absence of clear justification of the review. Indeed, the review started with very beginning information on origins of DNA damage but then includes parts, which describes only some features of BER, NER, MMR, HR, and NHEJ pathways. It will be very useful in Abstract and Introduction give clear aim of this manuscript and keep information focus within this stated aim. Current version partially looks like “patchwork quilt” when in mentioned pathways authors give some detail, but "forgot" others. For example, mechanism of BER is missed, different types of catalytic activities of DNA glycosylases are not mentioned, moreover, given text misleading the readers by statement that DNA glycosylases can only hydrolyses N-glycosidic bond (line 130), mechanism of damaged nucleotide search is not fully discussed for all pathways.
In fact, BER pathway is described very poor. Authors are focused on two DNA glycosylases in the NCP context. Why only OGG1 and UDG were selected, but other DNA glycosylases for example NTHL1, NEIL1, or AAG as well as AP endonuclease APE1 were missed?
Why only BER enzymes highlighted in the nucleosomal context, whereas other pathways did not at all?
The interplay between multiple BER enzymes is also not discussed. However, there are many examples of protein-protein interactions, damaged DNA passing, mutual effect of enzymes on their activity and coupled conformational changes of enzymes interacting with DNA.
It is not clear why authors schematically represent only NER, HR, and NHEJ, but not BER and MMR.
The role of PARP enzymes in DDR pathways is not fully discussed.
Descriptions of conformational changes of NCP or enzymes, which required for signal passing or formation catalytic complexes, are appeared sometimes in the text, but do not allow to draw common conclusion on the importance of such changes as finally stated in the Conclusion section.
Given that the authors try to make a case for the need for continued research of DNA repair mechanisms to improve medical treatments and advance drug discovery it is mandatory to mention current level of such therapeutics.
Minor
1) Lines 134-136, why only AP lesions in chromatin are mentioned?
2) Figure 7a does not contain pink colored structure as stated in the legend.
Reviewer 2 Report
Comments and Suggestions for Authors
The review titled "The repair pathways and mechanisms involve in the DNA damage response" summarizes well the current state of knowledge about DNA repair. The literature lacks a recent review of this type.
Several comments –
1. HR and NHEJ are the main repair mechanisms for DSBs, but there are other mechanisms. You should include alt-EJ and single-strand annealing repair pathways, or at least change "DSBs are repaired by HR or NHEJ..." to "DSBs are repaired mainly by HR and NHEJ...".
2. The phosphorylation sites of Ku70 and Ku80 were mentioned. Why not those of XRCC4 (Ser320) or Artemis (S516 and S645)?
3. Lines 499-501: Edit this sentence - it's unclear how Artimis is connected.
4. NHEJ is an error-prone repair mechanism, whereas HR is considered accurate. The first sentence of this paragraph should be edited so it will be clear that the sequence errors may occur in NHEJ and not in HR.
In addition, NHEJ is error-prone not only because it is faster than HR, but also because of its mode of action, which introduces small deletions and mutations.
5. There is no discussion of the choice of DSB repair pathways in the review. There is a lot of data regarding this in the literature and it should be included also in this review.
Minor comment-
1. In the review, ionizing radiation is mentioned three times, twice as its full name and once as IR. The abbreviation of IR should be written in the first time that it is mentions or be replaced by the pull length.
Round 2
Reviewer 1 Report
Comments and Suggestions for Authors
The manuscript could be accepted in present form.
Author Response
We thank the reviewer for their constructive feedback on our manuscript. Their suggestions helped to improve the focus and scope of our review.